# Accelerating Sparse Convolution with Column Vector-Wise Sparsity

**Yijun Tan**[1,2] **Kai Han**[3] **Kang Zhao**[3] **Xianzhi Yu**[3] **Zidong Du**[1] **Yunji Chen**[1,2] **Yunhe Wang**[3,*] **Jun Yao**[3]

[1]SKL of Processors, Institute of Computing Technology, CAS
[2]University of Chinese Academy of Sciences
[3]Huawei Noah's Ark Lab
tanyj1998@gmail.com, {kai.han,yunhe.wang}@huawei.com

## Abstract

Weight sparsity is a promising approach to reducing the model size and computation cost of convolutional neural networks (CNNs). Nevertheless, non-zero weights often distribute randomly in sparse CNN models, introducing enormous difficulty in obtaining actual speedup on common hardware (e.g., GPU) over their dense counterparts. Existing acceleration solutions either require hardware modifications for irregular memory access support or rely on a partially structured sparsity pattern. Neither of these methods is capable of achieving fruitful speedup on convolution layers.

In this work, we propose an algorithm-software co-designed sparse convolution based on a novel out-vector-wise (OVW) sparse pattern. Building on the insight that vertical vector integrity can preserve continuous memory access in IM2COL, the OVW pattern treats a $V \times 1$ vector as unit. To reduce the error caused by sparsity, we propose an equivalent transformation process, i.e., clustering-based channel permutation, to gather similar rows together. Experimental evaluations demonstrate that our method achieves a $1.7\times$ and $3.2\times$ speedup over the SOTA solution and the dense convolution of ResNet50 on NVIDIA V100 at 75% sparsity, respectively, with only negligible accuracy loss. Moreover, compared to the SOTA solution that achieves speedups only on data with 60% sparsity or more, our method begins to obtain speedups on data with only 10% sparsity.

## 1 Introduction

Recently, convolutional neural networks (CNNs) have yielded astonishing results in many important domains such as vision [8], and language [18]. With CNN algorithm developing rapidly, CNN models' storage and computing overhead grow exponentially. To significantly reduce both the computations and memory access, weight sparsity has been adopted as a promising approach to improve hardware efficiency.

Despite the success in reducing computations and data access, unconstrained, fine-grained sparsity fails to bring practical speedups on common GPUs. This is because unstructured sparsity generally induces tremendous access conflicts and load unbalances, which lowers GPU's performance. For example, on NVIDIA V100, the sparse matrix multiplication performs not faster than the dense matrix multiplication until the sparsity ratio is over 95% [17, 3].

Unfortunately, existing solutions either require hardware modifications or only partially address the problem by being constrained with structured, coarse-grained sparsity, resulting in high accuracy loss. The former is to leverage the sparse matrix-matrix multiplication (e.g., SPMM) operation

---

*Corresponding author

36th Conference on Neural Information Processing Systems (NeurIPS 2022).

on GPU. While directly applying SPMM on sparse CNNs can run even slower than dense CNNs [17],productive SPMM acceleration solutions[21, 14] often require their unique need for dedicated hardware support to overcome discontinuous memory access, which is impractical. The latter is to leverage the general matrix-matrix multiplication (e.g., GEMM) operation on GPU.

Recent works focus on structured sparsity with different sparse patterns to gain speedup benefits from weight sparsity. Block sparsity[4] manages to restore the spatial locality of matrices to a large extent, at the cost of a strict restriction on the non-zero Balanced sparsity[2, 19, 15], newly introduced on NVIDIA A100 GPU[14], however, lacks flexibility in choosing model sparsity rate that only exact 50% sparsity ratio could be deployed on this dedicated hardware. These efforts achieve some palpable acceleration compared to dense GEMM operation, but they all struggle to attain similar results on convolution layers which have proven to be a greater challenge.

To tackle these problems, here we present a novel sparse convolution acceleration algorithm featured with column-wise sparsity and implicit matrix multiplication. Specifically, the proposed column-wise sparsity is dubbed the out-vector-wise (OVW) sparse pattern since the pattern sparsifies a matrix by treating a V×1 vector as an entirety, as shown in Figure 1. During convolution, the OVW pattern can hold both strong memory consistency and high data reuse rates of input matrices using implicit matrix multiplication. Moreover, we propose to employ channel permutation and row clustering to improve the accuracy of OVW sparse pattern-based CNNs. Besides, a GPU kernel is carefully designed to ensure that our OVW sparse pattern is supported by common GPUs. With these efforts, our algorithm predominantly outperforms other sparse convolution acceleration algorithms on various CNN models. More importantly, our algorithm can achieve the acceleration of convolutions even with a very low weight sparsity ratio, e.g., 10%. Instead, prior arts can only work fine when the weight sparsity ratio is over 60%.

The main contributions of this paper are listed as follows:

- We propose a vector-based sparsity pattern, i.e., the OVW pattern to balance inference accuracy loss and computation efficiency in a hardware-friendly manner.

- We implement a new GPU convolution kernel to support the OVW pattern. The kernel utilizes the technique of extracting filter location information which can further reduce inference runtime.

- We propose a heuristic clustering method to obtain an appropriate channel permutation for reducing accuracy drop during weight pruning. This channel permutation operation is conducted offline, which does not affect inference time.

- Our GPU kernel can accelerate convolution at a wide range of model sparsity rates. With few accuracy loss, the kernel can speed up ResNet50 by $1.7\times$ and $3.2\times$, respectively over the SOTA solution and the dense cuDNN convolution on NVIDIA V100 GPU at 75% sparsity level.

## 2 Related work

### 2.1 Software-only Acceleration For Sparse CNN Model

Weight pruning has been a popular technique for efficient CNN inference. Early studies[7, 6] show that removing a large proportion of unimportant connections in CNN models does not necessarily lead to inference accuracy impairment. Reducing parameters helps exploit redundancy in CNN models, which requires fewer computations and data accesses. However, for CNN inference, weight pruned sparse CNNs usually perform worse than dense counterparts, unless the CNN sparsity ratios are substantial, i.e., very sparse CNNs.

To address this issue, methods other than unstructured sparsity are exploited. Researchers exploit various constraints on sparsity patterns in exchange for computation efficiency. A primary domain of this region is filter pruning, where parameters of an entire filter are pruned or kept as a whole. However, this direct modification of channel size suffers a sharp accuracy drop [10, 11, 16]. Moderate sparsity patterns are also examined, such as block sparsity[17], which is proposed to elevate the spatial locality of sparse matrices. But this feature can achieve speedup only when sparsity ratios are larger than 70%. Tile-wise sparsity[5] endows weight patterns with more flexibility. Compared to previous methods, balanced sparsity[15, 19] is more feasible with recent support from NVIDIA A100 GPU which

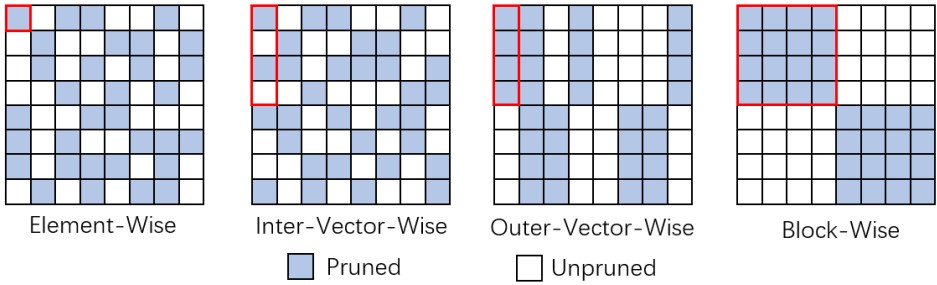

Figure 1: Comparison of four patterns with 50% sparsity.

directly optimizes 2:4 balanced sparsity. Recent work $Shfl\_BW$[9] uses matrix transformation to utilize block sparsity's computation efficiency while removing some of its constraints. In this way, the threshold of weight sparsity ratio that enables acceleration is reduced from 70% to 60%. Different from prior works which only work for very sparse matrices, in this paper, our algorithm can achieve speedup when the sparsity ratio is only 10%.

## 2.2 GEMM Based Convolution

GEMM has been adopted widely to perform convolution and it performs significantly better than other convolution methods such as FFT, and Winograd on modern commercial hardware accelerators such as TPUs and GPUs. The GEMM-based algorithms could be further divided into two types: explicit matrix multiplication and implicit matrix multiplication. Explicit matrix multiplication uses IM2COL to adapt inputs for GEMM. IM2COL is an IO-intensive operation, which brings in significant workload other than computation cost[1]. Implicit matrix multiplication merges these operations for more efficient memory accesses. It updates pointers of feature map in shared memory and performs tile-based matrix multiplication simultaneously. On NVIDIA V100 GPUs, explicit GEMM convolutions consume on average 120%, 126%, and 142% in time compared to implicit GEMM-based convolution on convolution layers of Alexnet, Resnet and Googlenet [20].

Yet, few studies have investigated the sparse convolution with implicit GEMM. Performing sparse convolution by GEMM is always through explicit matrix multiplication rather than implicit matrix multiplication. This is because of the IM2COL operation which is extremely difficult if not impossible for sparse matrix multiplication, as sparse matrices are compactly compressed and irregularly stored. As a result, implicit GEMM who does not have to suffer from the costly IM2COL operation has the potential to achieve higher efficiency for sprase convolution. In this paper, we investigate the implicit GEMM-based sparse convolution to leverage the high-performance GEMMs on existing hardware.

## 3 Accelerating Sparse Convolution

In this section, we introduce our proposed sparse convolution algorithm, including the OVW pattern of sparsity for the proposed sparse convolution, its advantage in convolution computation and our detailed implementation on GPU.

### 3.1 The OVW Pattern

The OVW pattern belongs to the vector-wise(VW) pattern which is one of the three different pattern categories of sparsity in matrix. As shown in Fig 1, the first sparsity pattern is the element-wise(EW) pattern, corresponding to unstructured pruning, which evaluates each parameter individually. Having imposed no constraint on pruning, this pattern succeeded at model flexibility but struggled at actual acceleration due to its irregular memory accesses. The second sparsity pattern is the VW pattern, which can be further divided into the inter-vector-wise(IVW) and the OVW pattern. They both treat a V×1 vector as an entirety while the IVW pattern prunes a certain proportion of weights inside each vector and the OVW pattern focuses on the entire vector of weights. The third pattern is the block-wise(BW) pattern, and its minimum pruning granularity is a V×V block. This pattern has the highest computation efficiency, but its inference accuracy loss is high as well. In this work, we

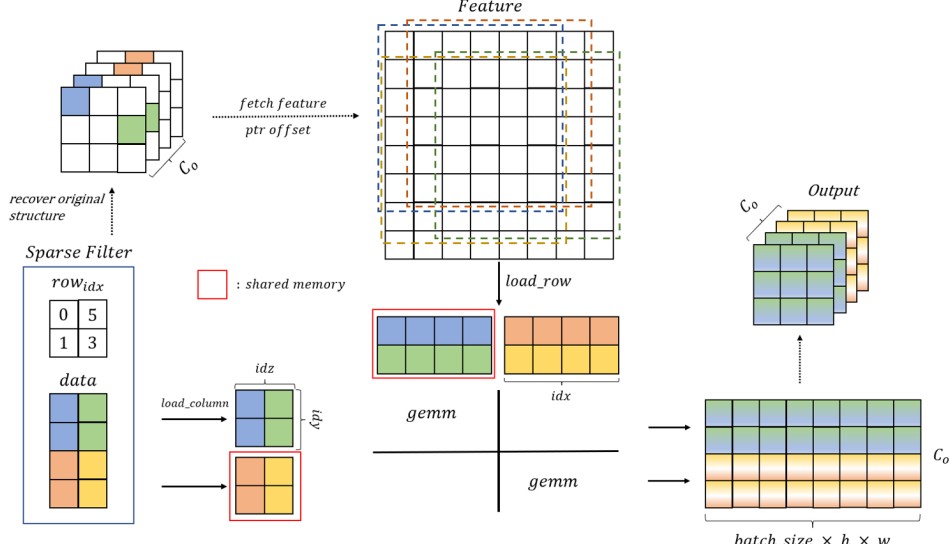

Figure 2: Illustration of GPU conv kernel. The solid lines are the actual computation flow and the dotted lines are processing pointers and descriptors.

use the OVW pattern since it shares the advantages of the VW pattern, which balances computation efficiency of BW and network accuracy of EW. The $Shfl\_BW$ pattern is actually a variant of this pattern, who uses an extra channel reordering procedure to gather block-wise pattern utilities.

## 3.2    The OVW Pattern's Advantage in Convolution Computation

The biggest advantage of the OVW pattern is that it fits the way an efficient dense warp-level GEMM instruction fetches input data. This instruction is the key contributor to most of the sparse matrix acceleration methods. The reasons are as followed.

As shown in Fig 2, the OVW pattern-based sparse convolution can be broken down into multiple dense matrix multiplications of smaller sizes. During the loading process of a dense convolution procedure, a column of filter data loaded into shared memory shares a specific position on the filter map. In the meantime, a continuous block in the feature map is loaded accordingly to prepare convolution. Several columns fetched from the filters together form the left input matrix of the block matrix multiplication and their corresponding feature data blocks form the right input matrix. Noticing that this forming process of input matrix does not require the loaded filter columns to be continuous, meaning that efficient dense operations can also be performed by grouping some unrelated columns. Based on this observation, we could take in multiple columns of irrelevant column indices from the OVW pattern sparse matrix and handle them in the same way as the dense GEMM operation. This similarity between our convolution algorithm and the implicit GEMM convolution guarantees us similar overall computation efficiency.

What's more, other sparsity such as the $Shfl\_BW$ pattern who is actually a variant of this pattern, uses an extra channel reordering procedure to gather block-wise pattern utilities. The OVW pattern, however, could be directly used in our convolution algorithm which denotes a higher acceleration potential. Compared to N:M sparsity,one of the IVW pattern, our approach does not need specialized hardware supports and it is much more elastic in selecting the sparsity ratio of each layer. Besides, the IVW pattern still faces the memory-bound issue, because the amount of redundant data that needs to be loaded into shared memory each time is equal to its sparsity ratio.

## 3.3    GPU Sparse Kernel Implementation

As shown in Algorithm 1, our convolution kernel implementation contains three steps. The first step is to get the corresponding feature pointer offsets through original filter structure information recovery. These parts of calculation are done by the function $Cal\_Thread\_Offset$. The second step is to

load data from both input matrices into shared memory. Some threads use the function $load\_column$ to load a column vector of length $TM$ from filters to shared memory with $DY$ threads. Threads then use the function $load\_row$ to load a row vector from the feature map in the same way. The third step calls warp matrix multiplication operators. When calculating matrix $A_{M \times K} \times B_{K \times N} = C_{M \times N}$, a three dimension parallelism$(DX, DY, DZ)$ thread is employed. $(DX, DY, DZ)$ threads loop along $(M, N, K)$ individually. Several threads together use the function $Warp\_MMA$ to multiply the loaded matrices and write back after results accumulation is finished. Each thread computes a tile matrix multiplication of the size $(TM, TN, TK)$.

The procedure of calculating the exact pointer offset of the feature map for corresponding filter columns contains two steps. During a convolution computation, the corresponding location of $a_{ij} \times b_{jk}$ is not obvious. Hence after loading $a_{ij}$ into shared memory, firstly, the GPU kernel has to fetch the column indices of $a_{ij}$ in the original filter. The column indices are then used to recover the exact position of this column in the filter map. Subsequently, the location offset of the corresponding data in the feature map is calculated, after which $b_{jk}$ can be finally located in the feature map.

---

**Algorithm 1:** Sparse convolution computation

**Data:** $row\_idx[]$, $filter[]$, $input[]$
**Result:** $output[]$

1   Shared memory $A[TM][TK]$, $B[TK][TN]$, $C[TM][TN]$;
2   **for** *Thread idx=1 **to** DX, idy=1 **to** DY, idz=1 **to** DZ* **do**
3      $offset$ = Cal_Thread_Offset($row\_idx[]$, $idx$, $idy$, $idz$);
4      **if** *idx < TK* **then**
5         Load_column($A$, $filter[idz][idx]$, $TM$, $idy$);
6      **end**
7      **if** *idy < TK* **then**
8         Load_row($B$, $input[offset]$, $TN$, $idx$);
9      **end**
10     Syncthreads();
11     Warp_MMA($A$, $B$, $idx$, $idy$);
12     Accumulate_Results($C$);
13     Store($output$, $C$);
14   **end**
15   Return output;

---

If only the column indices of sparse matrices are stored, their location information has to be recovered each time before the corresponding activation is loaded. Like the location offset in the feature map, the corresponding data location in the filter map could be prepared in advance. Because it is a constant for each thread during the whole process. Extra storage occupation of this technique is two extra dimension indices data array of filter map, which takes merely 3% total storage of a compressed model with vector length=64, and 6% with vector length=32, in exchange for 10% run time reduction of Resnet50's convolution layers on average. Considering that a sparse model is already highly compressed, this additional model redundancy is totally acceptable.

## 4   Pruning Algorithm

In this section, we introduce our pruning algorithm for the OVW pattern, including the channel permutation technique and our method to acquire a desired permutation order for it.

### 4.1   Channel Permutation

Our pruning method can be divided into two steps: shuffling the filter matrix rows in each layer and applying vector-wise pruning.

Here we will explain why filter permutation will do no harm to the network inference. Matrix multiplication only swaps the order in the output dimension and does not change the actual computation. Permuted operation results can be recovered through a reversed permutation of the operand output. As we only permute the output channel of each layer, the permuted order of the current layer will be

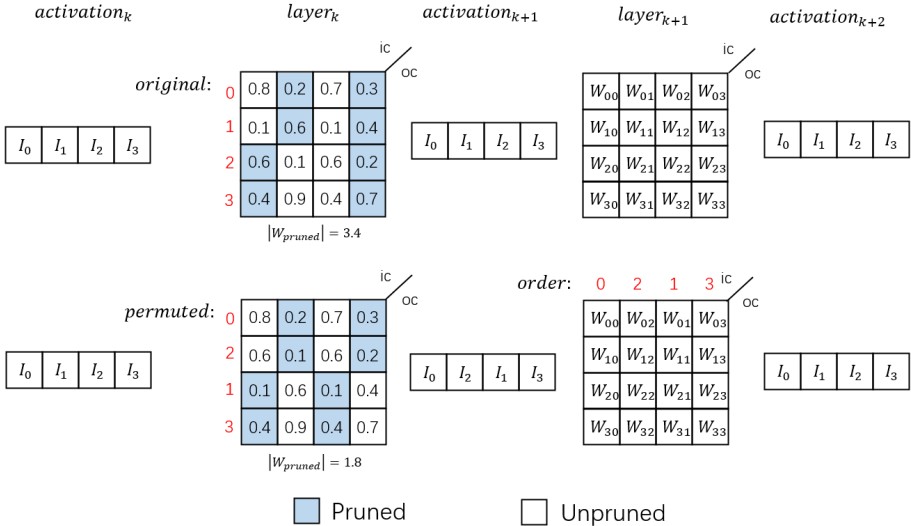

Figure 3: Illustration of the permutation transfer process. The permuted matrix kept a larger amount of absolute weight value after pruning at the same sparsity. Following activation is recovered by permuting the output channel of layer k and the input channel of layer k+1 simultaneously.

absorbed by the input channel dimension of the next GEMM-based layer(convolution or linear). Fig 3 shows one iteration for channel permutation between $layer_k$ and $layer_{k+1}$. As we have reordered the output channel of $layer_k$, the $activation_k$ is changed to the same order, but after we permute the input channel of $layer_{k+1}$, $activation_{k+1}$ is restored. More weights value could be saved after permutation. The same operation is then repeated on $layer_{k+1}$ and so on until every GEMM layer in the network is permuted. This permutation transfer procedure allows us to choose an appropriate permutation row order for every filter without altering the output of the network.

The rest of the layers such as the pooling layers and the activation layers involve no modification along channel dimension thus are not affected by this process. The BN layers and the bias added at the end of convolution layers and linear layers do not produce any new permutation order, but they have to permute according to the permutation passing through.

### 4.2 Row Clustering

---
**Algorithm 2:** Row clustering

**Data:** The original weight $W$, number of clusters $k$, number of selected column $m$.
**Result:** The reordered weight $RW$.
1 $RW = empty$;
2 **while** $W$ *is not empty* **do**
3      Sort the columns of $W$ by column variance;
4      Build $SampleW$ by selecting columns with the top-$m$ largest variances;
5      Get the $k$ clustered groups $G$ = Balanced_kmeans($SampleW$, $k$);
6      Select the group $g$ with maximum sum;
7      Append $g$ to $RW$;
8      Remove $g$ from $W$;
9 **end**
10 Return $RW$;

---

We use a row clustering method to obtain an appropriate permutation order. A heuristic indicator evaluating the quality of a permutation is the sum of absolute weight value being pruned, assuming that the method that the preservation of more important weights corresponds to less inference accuracy loss. An obvious route is that the weight rows with shorter distances are assigned to the same group as

Table 1: Network accuracy on Cifar100. V is the vector length of the OVW pattern. V for Vgg19 and Resnet is set to 64 for all layers. For SqueezeNet and MobileNetv2, the V value in the table is our average vector length of the whole network as we select an optimal V for each layer.

| Network | | Vgg19 | Resnet18 | Resnet50 | SqueezeNet | MobileNetv2 |
|---|---|---|---|---|---|---|
| Baseline | dense | 71.41 | 77.19 | 78.60 | 71.01 | 68.99 |
| Unstructured | | 71.62($\pm$0.11) | 76.42($\pm$0.04) | 77.99($\pm$0.08) | 70.27($\pm$0.15) | 68.89($\pm$0.06) |
| OVW | 80% | 71.36($\pm$0.04) | 73.67($\pm$0.35) | 75.80($\pm$0.16) | 69.05($\pm$0.12) | 68.31($\pm$0.11) |
| **OVW permuted** | | 71.46($\pm$0.06) | 74.23($\pm$0.22) | 75.99($\pm$0.18) | 69.29($\pm$0.15) | 68.52($\pm$0.05) |
| $\Delta$ | | +0.10 | +0.56 | +0.19 | +0.24 | +0.21 |
| $\bar{V}$ | | 64 | 64 | 64 | 44.48 | 32.02 |
| Unstructured | | 71.35($\pm$0.05) | 74.87($\pm$0.07) | 78.02($\pm$0.04) | 70.29($\pm$0.16) | 68.40($\pm$0.19) |
| OVW | 90% | 71.29($\pm$0.03) | 71.14($\pm$0.10) | 72.72($\pm$0.13) | 65.69($\pm$0.59) | 66.07($\pm$0.64) |
| **OVW permuted** | | 71.37($\pm$0.19) | 72.25($\pm$0.15) | 73.26($\pm$0.22) | 65.80($\pm$0.73) | 67.46($\pm$1.03) |
| $\Delta$ | | +0.08 | +1.11 | +0.54 | +0.11 | +1.39 |
| $\bar{V}$ | | 64 | 64 | 64 | 44.48 | 32.02 |

much as possible. $Shfl\_BW$ chooses the kmeans method for clustering, but after careful experiment, we found that the kmeans method does not suit this issue well. For starters, the number of elements in each group is set to be a fixed value (vector length), and kmeans requires additional operations to meet this demand. Also, the data dimension(input channel multiplies filter height and width) is too large, while the amount of data and groups is relatively small. Kmeans falls in local minima easily and the output cluster is extremely unstable. We introduced the balanced kmeans[13] to solve it and modified it to palliate both symptoms mentioned above.

Algorithm 2 shows the key steps of our algorithm. First, we construct a characteristic matrix by assembling rows with the highest variance, then cluster rows of this matrix to alleviate excessive dimension. We utilize balanced kmeans to get equal size clusters. In each iteration of balanced kmeans, instead of assigning each data vector to its nearest cluster center as the origin kmeans algorithm, a distance matrix between all the vectors and the current cluster centers is formed. We minimize the sum of distance while each cluster contains the same amount of data vector. This minimization problem can be converted to bipartite matching and we employ the Kuhn-Munkres algorithm to solve it. Secondly, for each clustering result, we only adopt the most important group under this feature to increase operation stability. This group is removed from the original matrix and then the feature matrix is reconstructed and clustered. Repeat the above steps until all data grouping completes, the permuted matrix and its permuting order is obtained.

## 5 Evaluation

### 5.1 Model Accuracy

We evaluate our method on several popular CNN models on NVIDIA V100 GPU. We only calculate the speedup of the convolution layers in the following results. Table 1 shows the accuracy of our method compared to unstructured sparsity where V stands for vector length. "OVW permuted" shows a better accuracy over "OVW non-permuted" on all CNN models.

The upper bound of V in our kernel implementation is 64 and we tend to make it as large as possible to maximize shared memory usage. However, group convolution has no filter data reusage for vector length larger than its group size. Similarly, V is set to 1 in depthwise convolution layers. Besides that, the first convolution layer of SqueezeNet has only 96 output channels. If the vector length is 64, the second tile in the output channel dimension only processes 32 rows which is only half loaded. In this case, vector length is set to 32 to maximize computation resource utilization. The V selection strategy here is to maximize V while utilizing shared memory resources as much as possible.

All the results in this table use the same fine-tuning process. We fine-tune each network for 40 epochs after pruning with the same learning rate of 0.0008. Also, each layer can hold different sparsity ratio credit to our acceleration for convolution at low sparsity ratio.

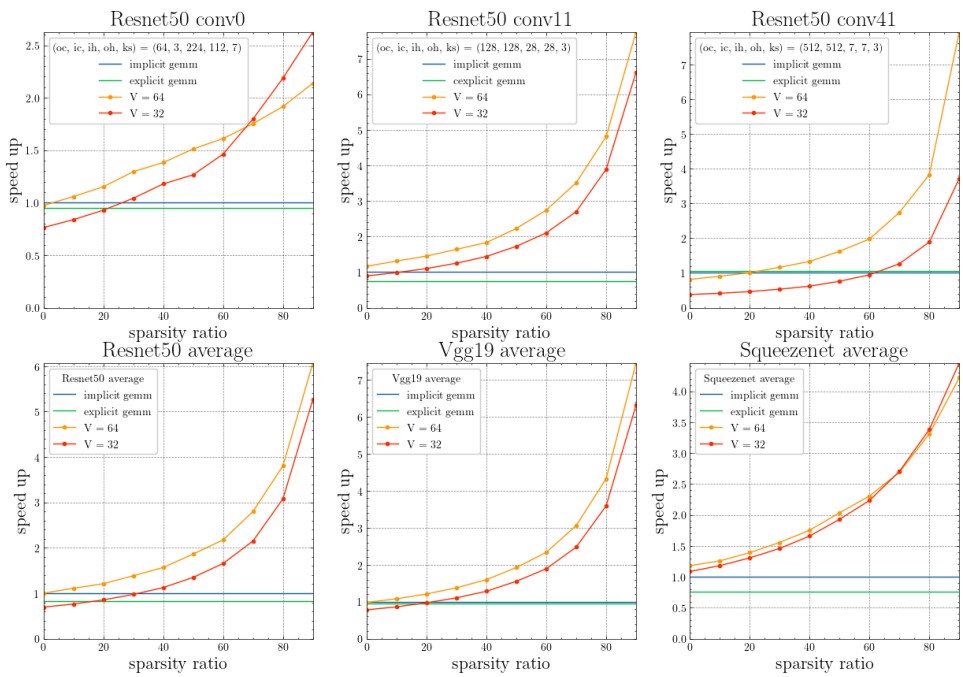

Figure 4: Speed comparison between our sparsity convolution kernel and two GEMM-based convolution in cuDNN library on NVIDIA V100 GPU.

Table 2: Comparing sparsity patterns on Resnet50 for ImageNet classification. Speedup of unstructured spasity is the same as dense because the fastest way to run it is to invoke the dense convolution kernel.

| Network | Sparsity ratio | Accuracy | Speedup |
|---------|----------------|----------|---------|
| Baseline | Dense | 76.12 | $1\times$ |
| Balanced sparsity (2:4)[15] | 50% | 76.29 | $1.3\times$ |
| $Shfl\_BW$[9] | 90% | 73.09 | $2.5\times$ |
| OVW | 30% | 76.17 | $1.38\times$ |
| OVW | 50% | 75.76 | $1.86\times$ |
| OVW | 70% | 73.35 | $2.79\times$ |

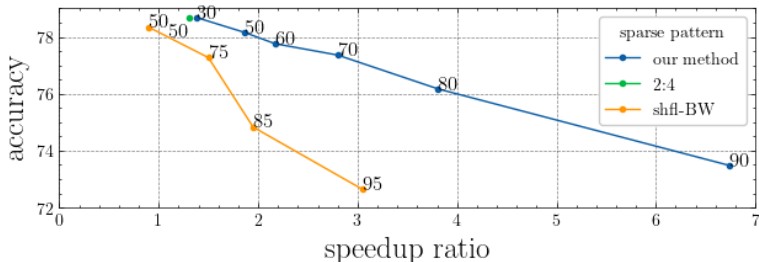

Figure 5: Speed up against the dense baseline on Resnet50 Cifar100 comparing to other vector-level sparse patterns on NVIDIA V100 GPU.

## 5.2 Convolution Kernel Speedup

As shown in Fig 4, we evaluate the speedup of our method on three popular CNN models. We use the cuDNN convolution operator as the dense baseline. The first three graphs in Fig 4 represent three typical convolution shapes in CNN models: small channel size with a large feature map, medium channel size with a medium feature map, and large channel size with a small feature map. Our kernel can accelerate all these types of convolution layers while exceeds at the twelfth convolution layer of Resnet50 which is the most used kind of convolution layer $4.8\times$ and $3.87\times$ over cuDNN on V100 at 80% sparsity, vector lengths 64 and 32.

## 5.3 Comparing Different Sparsity Patterns

We replicate two vector-level sparsity pattern, balanced sparsity(NVIDIA 2:4)[15] and $Shfl\_BW$ [9] for comparison. Other vector-level sparsity patterns such as Tile-wise[5] are slower than the former two patterns. Also, these patterns lack implementation for convolution.

Table 2 shows results of Resnet50 on ImageNet directly copied from $Shfl\_BW$ paper, where an expensive method Grow and Prune[12] is used to recover its accuracy. Grow and Prune is a sparsity pattern independent method. We fine-tune our pretrained Resnet50 model for 20 epochs with the learning rate of 0.001. We lower the sparsity of our network to 70% where our method demonstrates 73.35% top1 accuracy and $2.79\times$ speedup. Our method exhibits an obviously better speed-accuracy tradeoff compared to the Shfl_BW. The OVW pattern also achieves a better speedup compared to balanced sparsity while recovering the full accuracy of the original model.

To ensure fair comparison, we reproduce these results under the same setting on Resnet50 and Cifar100, as shown in Fig 5. We fine-tune each network for 40 epochs after pruning from pretrained dense models with the same learning rate of 0.0008. The OVW pattern dominates the speed-accuracy trade-off in vector-level sparsity.

# 6 Conclusion

Accelerating sparse convolution poses a greater challenge than accelerating sparse matrix multiplication. In this work, we propose a novel sparsity pattern, the OVW pattern to facilitate the sparse convolution acceleration under intact accuracy. Although, limitations exist that our method relies heavily on the hardware supports for the implicit GEMM convolution algorithm. The performance of our base dense kernel against the commodity unpublished ones is unstable. Our method does not acquire the same amount of acceleration rate on matrix multiplication and the acceleration rate of our method is subject to filter shape. Its performance also degrades in specialized convolution layers where data reusage opportunity is limited. In consideration of this, our GPU implementation still largely outperforms all sparse acceleration approaches exceedingly with a sparse pattern of similar flexibility. We hope this work can fill the vacancy of specialized sparse convolution kernel design and our methodology can inspire further research in this domain.

## Acknowledgement

We gratefully acknowledge the support of MindSpore, CANN (Compute Architecture for Neural Networks) and Ascend AI Processor used for this research.

This work is partially supported by the National Key R&D Program of China(under Grant 2017YFA0700902), the NSF of China(under Grants 61925208, 61732020, U19B2019), Strategic Priority Research Program of Chinese Academy of Science (XDB32050200), Beijing Academy of Artificial Intelligence (BAAI), CAS Project for Young Scientists in Basic Research(YSBR-029), Youth Innovation Promotion Association CAS and Xplore Prize.

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
