Table 1: Comparing with channel pruning. Speedups for HRank[11] are inferred from its theoretical FLOPs reduction.

| Network | Sparsity ratio | Accuracy | Speedup |
|---------|----------------|----------|---------|
| Baseline | Dense | 76.12 | $1\times$ |
| HRank[11] | 37% | 74.98 | $1.77\times$ |
| OVW | 50% | 75.76 | $1.86\times$ |
| HRank[11] | 46% | 71.98 | $2.63\times$ |
| OVW | 70% | 73.35 | $2.79\times$ |

Table 2: Comparing speedup with different vector lengths.

| Resnet50 layer | | 64 | 32 | 16 |
|----------------|-----|------|------|------|
| Conv0 | 50% | **1.51** | 1.27 | 1.12 |
| | 90% | 2.14 | **2.76** | 1.93 |
| Conv11 | 50% | **2.23** | 1.73 | 0.33 |
| | 90% | **7.75** | 6.62 | 1.38 |
| Conv41 | 50% | **1.62** | 0.76 | 0.31 |
| | 90% | **7.93** | 3.72 | 1.53 |

# A  Appendix

## A.1  Ablation Study

Table 1 shows the accuracy of our method compared to advanced channel pruning methods, HRank [11]. Channel pruning requires specialized training from scratch methods to recover its enormous accuracy drop. The OVW pattern demonstrates a better accuracy-speed tradeoff against it.

Table 2 supports our arguments for V that only vector lengths as large as 32 or 64 can minimize convolution kernel runtime.

Table 2 justified our system is robust on different types of GPU as long as its architecture supports optimization for the implicit GEMM convolution algorithm.

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

Table 3: Comparing speedup on RTX 3070.

| Resnet50 layer | | V100 | | RTX 3070 | |
|----------------|-----|------|------|------|------|
| | | V=64 | V=32 | V=64 | V=32 |
| Conv11 | 50% | 2.23 | 1.73 | 1.71 | 1.55 |
| | 90% | 7.75 | 6.62 | 6.71 | 6.39 |
| Conv41 | 50% | 1.62 | 0.76 | 1.30 | 0.68 |
| | 90% | 7.93 | 3.72 | 6.30 | 3.31 |