# OpenReview forum: "Accelerating Sparse Convolution with Column Vector-Wise Sparsity"
_NeurIPS.cc/2022/Conference — NeurIPS 2022 Accept_

### Official Review · Reviewer_m2da · 2022-07-04

**Rating:** 7
**Confidence:** 4
**Soundness:** 3 good
**Presentation:** 3 good
**Contribution:** 3 good

**Summary:**

This paper proposes an out-vector-wise (OVW) pattern-based sparse convolution for accelerating neural networks. The proposed OVW pattern can preserve continuous memory access. Moreover, the clustering-based channel permutation method is introduced to reduce error caused by sparsity. The sparse convolution is efficiently implemented using CUDA for GPU acceleration. The experimental results demonstrate the efficiency of the proposed method over unstructured sparsity.

**Questions:**

please refer to the weaknesses.

**Limitations:**

Yes

**Strengths And Weaknesses:**

Strengths:

- The proposed out-vector-wise (OVW) sparse pattern can preserve continuous memory access which is beneficial to GPU acceleration, while unstructured sparsity involves random memory access which is unfriendly for hardware.
- Sparse model acceleration is an important problem for mobile devices. The proposed method achieves a better tradeoff between inference speed and accuracy over unstructured sparsity and channel sparsity.
- The clustering-based channel permutation method can reduce error caused by sparsity.

Weaknesses:

- The paper claims that the OVW pattern can achieve a better tradeoff than channel pruning. please add the experimental comparison to evaluate this claim.
- The hyperparameter V determines the length of a vector. How does it affect the inference speed? It would be better to present an ablation study of it.

---

> ### Author Response · Authors · 2022-08-02
> **Response to Reviewer m2da**
>
> We thank the reviewer for the constructive feedback.
>
> >The paper claims that the OVW pattern can achieve a better tradeoff than channel pruning. please add the experimental comparison to evaluate this claim.
>
> Inference accuracy for channel pruning generally drops too far which has to be saved by complicated training methods while the accuracy of our method is finetuned from pretrained models.
> We add an ablation study with an advanced channel pruning method HRank on Resnet50 imagenet dataset. These results can demonstrate a better accuracy-speed trade-off.
>
> **Table 3**
>
> | Network | sparsity | accuracy | speedup |
> | ---- | ---- | ---- | ---- |
> |	Baseline                   | Dense  |  76.12   | $1\times$   |
> |	HRank                      | 37\%   |  74.98   | $1.77\times$|
> |	OVW                        | 50\%   |  75.76   | $1.86\times$|
> |	HRank                      | 46\%   |  71.98   | $2.63\times$|
> |	OVW                        | 70\%   |  73.35   | $2.79\times$|
>
> >The hyperparameter V determines the length of a vector. How does it affect the inference speed? It would be better to present an ablation study of it.
>
> V is restricted by the kernel implementation. Vector lengths as large as 32 or 64 can minimize convolution kernel runtime.
> We add an ablation study for V=16. These results can justify our argument for V.
>
> **Table 4**
>
> |Resnet50 layer | sparsity | 64 | 32 | 16|
> | ---- | ---- | ---- | ---- | ---- |
> | conv0 |50\% | 1.51 | 1.27 | 1.12 |
> | conv0 |90\% | 2.14 | 2.76 | 1.93 |
> | conv11|50\% | 2.23 | 1.73 | 0.33 |
> | conv11|90\% | 7.75 | 6.62 | 1.38 |
> | conv41|50\% | 1.62 | 0.76 | 0.31 |
> | conv41|90\% | 7.93 | 3.72 | 1.53 |
>
> We also add the detail of our V selection strategy in Section 5.1.
> “The upper bound of V in our kernel implementation is 64 and we tend to make it as large as possible to maximize shared memory usage. However, group convolution has no filter data reusage for vector length larger than its group size. Similarly, V is set to 1 in depthwise convolution layers. Besides that, the first convolution layer of SqueezeNet has only 96 output channels. If the vector length is 64, the second tile in the output channel dimension only processes 32 rows which is only half loaded. In this case, vector length is set to 32 to maximize computation resource utilization. The V selection strategy here is to maximize V while utilizing shared memory resources as much as possible.”

---

> > ### Comment · Reviewer_m2da · 2022-08-09
> > **After response**
> >
> > Thanks for your response. My comments are well addressed in this response, especially on experiments. Thus, I keep my original rate to accept this paper.

---

### Official Review · Reviewer_HQDT · 2022-07-11

**Rating:** 7
**Confidence:** 4
**Soundness:** 3 good
**Presentation:** 3 good
**Contribution:** 3 good

**Summary:**

This paper aims to speed up sparse convolutional neural networks on GPU devices. Instead of unstructured sparsity pattern, this paper proposes a vector-based sparsity pattern (OVW) to balance accuracy loss and computation efficiency. Based on the OVW pattern, a heuristic clustering method is introduced to obtain an appropriate channel permutation with smaller accuracy loss. Moreover, an efficient CUDA kernel is implemented for OVW-based sparse convolution. Extensive experiments on several network architectures demonstrate the efficiency of the method.

**Questions:**

- If I understand correctly, the hyperparameter $V$ should be an integer. In table 1, why $V$ is 44.48 (32.02) in the second last (last) row?
- The proposed OVW sparsity pattern is faster than unstructured sparsity. It would be better to include the comparison with unstructured sparsity in Figure 5.


**Limitations:**

Yes.

**Strengths And Weaknesses:**

**Strengths:**
- This work is clearly motivated. The unstructured sparsity is hard to be accelerated on GPUs, while channel-wise sparsity tends to have a large accuracy drop. The sparsity pattern between these two would have a better trade-off.
- The proposed vector-based sparsity pattern (OVW) is reasonable and technically sound. It can preserve continuous memory access for efficient GPU implementation.
- Extensive evaluation and impressive results. This paper demonstrates the power of the proposed sparse convolution on different network architectures, outperforming representative unstructured sparse methods.

**Weaknesses:**
- The proposed method can achieve practical speedup on V100 GPU. It would be better to verify on more GPU types.
- The writing should be improved. There are some typos:
Common issue: there should be a space before left bracket.
Line34: Missing citation number.
Line270: Fig5 -> Fig 5.

---

> ### Author Response · Authors · 2022-08-02
> **Response to Reviewer HQDT**
>
> We thank the reviewer for the constructive feedback.
>
> >The proposed method can achieve practical speedup on V100 GPU. It would be better to verify on more GPU types
>
> Generally, our method can be used on any GPU that supports the implicit GEMM convolution algorithm.
> We add an evaluation on more GPU types in the ablation study. These results justified our system is robust.
>
> **Table 5**
>
> |Resnet50 layer | | V100 | | RTX 3070 | |
> | ---- | ---- | ---- | ---- | ---- | ---- |
> | | | V=64 | V=32 | V=64 | V=32|
> |Conv11 | 50\% | 2.23 | 1.73 | 1.71 | 1.55 |
> |conv11 | 90\% | 7.75 | 6.62 | 6.71 | 6.39 |
> |Conv41 | 50\% | 1.62 | 0.76 | 1.30 | 0.68 |
> |conv41 | 90\% | 7.93 | 3.72 | 6.30 | 3.31 |
>
> >The writing should be improved. There are some typos: Common issue: there should be a space before left bracket. Line34: Missing citation number. Line270: Fig5 -> Fig 5
>
> We have revised the typo in fig 5 and checked the rest of this paper to ensure no more typos.
>
> >If I understand correctly, the hyper parameter  should be an integer. In table 1, why  is 44.48 (32.02) in the second last (last) row?
>
> We revise the explanation for “V” in Table 1 caption.
> “V is the vector length of the OVW pattern. V for Vgg19 and Resnet is set to 64 for all layers. For SqueezeNet and MobileNetv2, the V value in the table is our average vector length of the whole network as we select an optimal V for each layer.”
>
>
> We also add the detailed argument of our V selection strategy in Section 5.1.
> “The upper bound of V in our kernel implementation is 64 and we tend to make it as large as possible to maximize shared memory usage. However, group convolution has no filter data reusage for vector length larger than its group size. Similarly, V is set to 1 in depthwise convolution layers. Besides that, the first convolution layer of SqueezeNet has only 96 output channels. If the vector length is 64, the second tile in the output channel dimension only processes 32 rows which is only half loaded. In this case, vector length is set to 32 to maximize computation resource utilization. The V selection strategy here is to maximize V while utilizing shared memory resources as much as possible.”
>
>
> >The proposed OVW sparsity pattern is faster than unstructured sparsity. It would be better to include the comparison with unstructured sparsity in Figure 5
>
> As we stated in the background, only with over 95% sparsity ratio or more, the unstructured sparse matrix multiplication can achieve the same performance as the dense matrix multiplication. So the fastest way to run an unstructured sparse network is to use the dense convolution kernels which is equivalent to the baseline line of Table 2 and Fig 5.
> We add an illustration for this in Section 5.1.

---

### Official Review · Reviewer_C8Jt · 2022-07-12

**Rating:** 4
**Confidence:** 4
**Soundness:** 3 good
**Presentation:** 1 poor
**Contribution:** 3 good

**Summary:**

This paper introduces a new sparse convolution kernel, with the following properties:

* The filters are "outer-vector-wise" sparse. This means that the matrix of flattened filters has its columns split into vectors, and each vector is either zero or non-zero.
* This allows the kernel to use implicit matrix multiplication. (In explicit matrix multiplication the `im2col` operation is used to extract patches of the features map, which are then used in a batched matrix multiply. In implicit matrix multiplication, pointers into the feature map are used instead of explicitly extracting the patches. This is faster.)

When compressing a dense network, there are two ways that the sparsity pattern can be optimized:

* The sparsity pattern can be chosen (i.e., which vectors are zero and non-zero, under the constraint that each column of the flattened filter matrix must have the same number)
* The rows of the flattened filter matrix can be permuted (i.e., the output channels can be permuted), since this doesn't actually change the computation (as long as the next layer's input channels are permuted correspondingly).

The authors propose a row clustering algorithm which uses k-means to find a permutation of the rows that maximizes the weights of the unpruned entries.

The results show that this sparsity kernel can reach significant speedups at relatively low sparsity levels. The experiments also show that the test accuracy remains relatively high after pruning and fine-tuning. Lastly, it's shown that the permutations provide a small but consistent improvement in the test accuracy compared to not permuting.

**Questions:**

**Writing**

Overall the text requires a significant amount of editing. For example, the abstract contains typos ("re[p]ly on") and grammar mistakes such as incorrect plurals/singulars ("pattern[s]", "speedup[s]"), missing articles ("[the] OVW pattern"), and incorrect conjugations ("achieve[s]"). I found the writing to be meandering and unstructured throughout. As a reader it was often unclear to me what the main point of each section was. I suggest the authors spend some time editing the text for both grammar and structure.

The text could also be more precise to help ensure reproducibility. Section 4.2 is a particular example of this, with vagues statements ("kmeans requires additional operations to meet this demand") and a lack of precision, e.g., "We employ Kuhn-Munkres algorithm to solve the distance matrix and optimize the mean square error of it". Which distance matrix? Mean square error of what? None of this seems to appear in algorithm 2.

**Illustrations**

I believe figure 1's illustration of "inter-vector-wise" sparsity contains a small mistake (the two top right vectors have 3 instead of 2 unpruned entries)

Figure 2 is confusing to me: The diagram makes it look as if the sparse filter is first densified to then be passed to `load_columns`, the output of which seems the same as the original input `data`? There's a function `recover` which seems to just be `reshape`? And the figure suggests that the feature offsets are a function of the filters' values, whereas they are actually computed based on the row indices (`row_idx`).

**Experimental results**

I believe it would be valuable to perform multiple runs and provide confidence intervals on the presented results, given the small margins (e.g., 65.49 and 65.46 for SqueezeNet OVW permuted/non-permuted).

The "V" in the table of results should probably be explained in the caption. I assume it's the vector length as in the text, but what does it mean for the vector length to be 44.48 or 32.02? And perhaps it is better to move this column to the end of the table, to make it easier to compare the OVW columns to the baseline columns.

The results are also a bit hard to parse since they require the reader to compare numbers across 6 columns to see which ones are bigger or smaller. Perhaps the authors could report "drop/increase in accuracy compared to baseline" instead? And perhaps the columns are better grouped by their sparsity level rather than by the sparsity type.

**Training**

This paper assumes a setting of dense training followed by compression for deployment. I would be interested to see the authors discuss using these sparse kernels during training as well.

**Title**

The title of the paper is overly general. Many papers discuss accelerating sparse convolutions after all. Perhaps something along the lines of "Column-wise sparsity and implicit matrix multiplication for efficient sparse convolutions" would be more descriptive of the content of the paper.

**Limitations:**

The authors claim to have discussed the limitations of their method, but it's not clear to me where they did so.

**Strengths And Weaknesses:**

I think this paper proposes a very neat sparse convolutional kernel. The outer-vector-wise sparsity seems like a good trade-off between fast structured sparsity (e.g., block-wise sparsity) and flexibility (e.g., unstructured sparsity). This works nicely with the implicit matrix multiplication approach, which gives impressive experimental results.

The main weakness of this paper is the presentation and reproducibility. The writing is not good, which makes it more difficult to follow along than it should be, and the pseudo-algorithms seem to be missing important details required to reproduce their results. Similarly, the figures are not as elucidating as they could be, and the experimental results are presented in a way that is difficult to parse.

All in all, the amount of editing that needs to be done to bring this paper to a publishable state is on the borderline of what is reasonable to expect from a conference rebuttal cycle. This makes it difficult to recommend this paper as is.

---

> ### Author Response · Authors · 2022-08-02
> **Response to Reviewer C8Jt part2**
>
>
> **Experimental results**
> > I believe it would be valuable to perform multiple runs and provide confidence intervals on the presented results, given the small margins (e.g., 65.49 and 65.46 for SqueezeNet OVW permuted/non-permuted).
>
> We add more runs to it and updates the highest score for all network in the table. Table 1 is redrawn to improve readability.
> The specific accuracy intervals of Squeezenet are listed as below.
>
> | sparsity | unstructured | OVW | OVW permuted |
> | ---- | ---- | ---- | ---- |
> | 80 | 70.27 ($\pm$0.15)  | 69.05 ($\pm$0.12) | 69.29 ($\pm$0.15) |
> | 90 | 70.29 ($\pm$0.16)  | 65.69 ($\pm$0.59) | 65.80 ($\pm$0.73) |
>
>
> > The "V" in the table of results should probably be explained in the caption.
>
> We add explanation for “V” in Table 1 caption.
> “V is the vector length of the OVW pattern. V for Vgg19 and Resnet is set to 64 for all layers. For SqueezeNet and MobileNetv2, the V value in the table is our average vector length of the whole network as we select an optimal V for each layer.”
>
> We also add the detail of our V selection strategy in Section 5.1.
> “The upper bound of V in our kernel implementation is 64 and we tend to make it as large as possible to maximize shared memory usage. However, group convolution has no filter data reusage for vector length larger than its group size. Similarly, V is set to 1 in depthwise convolution layers. Besides that, the first convolution layer of SqueezeNet has only 96 output channels. If the vector length is 64, the second tile in the output channel dimension only processes 32 rows which is only half loaded. In this case, vector length is set to 32 to maximize computation resource utilization. The V selection strategy here is to maximize V while utilizing shared memory resources as much as possible.”
>
>
> > The results are also a bit hard to parse
>
> We reorganize the presentation of Table 1 as follows. The results are grouped by their sparsity level. Note that the speed of the unstructured sparse model is similar to the dense model, which is much slower than ours (details see Figure 4 and Figure 5 in the manuscript).
>
> *Table 1*
>
> | Network | sparsity | Vgg19 | Resnet18 | Resnet50 | SqueezeNet | MobileNet v2 |
> | ---- | ---- | ---- | ---- | ---- | ---- | ---- |
> | Baseline  | dense | 71.41 | 77.19 | 78.60 | 71.01 | 68.99 |
> | Unstructured | 80 | 71.73 | 76.46 | 78.07 | 70.42 | 68.95 |
> | OVW          | 80 | 71.39 | 74.02 | 75.96 | 69.18 | 68.43 |
> | OVW permuted | 80 | 71.52 | 74.45 | 76.17 | 69.41 | 68.56 |
> | $\Delta$   | | +0.13 | +0.43 | +0.20 | +0.23 | +0.13 |
> | V  | | 64 | 64 | 64 | 44.48 | 32.02 |
> | Unstructured | 90 | 71.40 | 74.95 | 78.02 | 70.45 | 68.59 |
> | OVW          | 90 | 71.32 | 71.24 | 72.85 | 66.28 | 66.71 |
> | OVW permuted | 90 |71.57 | 72.40 | 73.48 | 66.54 | 68.42 |
> | $\Delta$   | | +0.25 | +1.16 | +0.63 | +0.26 | +1.71 |
> | V  | | 64 | 64 | 64 | 44.48 | 32.02 |
>
>
>
>
> **Training**
> > I would be interested to see the authors discuss using these sparse kernels during training as well.
>
> The target of our work is to accelerate convolution during inference not training. As for training, applying our method in the backpropagation process will cause the input channel dimension be restricted by V. This will further limit the sparsity structure of the filter. Thus, our method cannot be directly used for accelerate training process.
>
> **Title**
> >The title of the paper is overly general. Many papers discuss accelerating sparse convolutions after all.
>
> We agreed that the title could be overly general. So we changed it to “Accelerating Sparse Convolution with Column Vector Wise Sparsity”. We stick to the point of view that our work is the first to directly optimize sparse convolution itself without drawing any connection to sparse matrix multiplication methods.
>
> **Limitations**
> We add more detailed limitation descriptions in the conclusion section.
> The limitations of our work are listed as follows:
> + Our method relies heavily on the hardware supports for the implicit GEMM convolution algorithm. The performance of our base dense kernel against the commodity unpublished ones is unstable.
> + Our method does not acquire the same amount of acceleration rate on matrix multiplication. Because the latency of data pointer offset recovery from sparse filter data to dense filter data is hidden in the implicit GEMM convolution algorithm while it is entirely opposed in a matrix multiplication process.
> + The acceleration rate of our method is subject to filter shape. Its performance also degrades in specialized convolution layers where data reusage opportunity is limited.

---

> > ### Comment · Reviewer_C8Jt · 2022-08-10
> > **Response**
> >
> > Thank you for your response. I've also read the other reviews and responses and have been impressed by the additional experiments that the authors have performed in order to address the raised weaknesses.
> >
> > **Writing**
> >
> > I believe the edits have increased the readability of the paper. However, significant editing for language still remains. For example, going over the first 4 paragraphs alone I already came across the following phrases which are odd or incorrect. Since one of the main goals of a publication is to convey research with clarity and precision, I think this still weighs against acceptance.
> >
> > * 12: treats [...] as a~n entirety~ *unit*
> > * 23-24: reduce both computation~s~ and ~data~ *memory* access~es~
> > * 24: sparsity has been ~taken~ *adopted* as a promising approach
> > * 26: ~the~ [...] sparsity fails to bring
> > * 29: under the sizes (?)
> > * 32-33: ~high~ *low* accuracy ~loss~
> > * 33: the formal (?) is to leverage
> >
> > Also, in table 1 the header "sparsity" is missing, so it seems as if the networks are 80%/90% dense rather than 80%/90% sparse.
> >
> > **Experimental results**
> >
> > It it is not good practice to only report the run with the highest accuracy in the text. This favours methods with high variance and a large amount of runs, and it does not give the reader an accurate picture of the numbers they can expect when reproducing your results. Please include the average accuracies along with their confidence intervals in the table instead.
> >
> > I believe that if these two changes are made (editing for language/grammar, and reporting confidence intervals in the main text) I would be willing to increase my score to accept.

---

> > > ### Author Response · Authors · 2022-08-10
> > > **Response to reviewer C8Jt:**
> > >
> > > Thanks again for your constructive feedback. We appreciate your efforts in helping us improve our paper.
> > >
> > > **Writing**
> > >
> > > We have revised those incorrect or inappropriate phrases you addressed and fixed the missing header in Table 1.
> > > We also checked and edited the rest of this paper carefully.
> > >
> > > **Experiment results**
> > >
> > > We agreed that average accuracy and confidence intervals could better improve reproductivity.
> > > We have added them in table 1. It will be shown in the final version of this paper if accepted.
> > >
> > >
> > > *Table 1*
> > >
> > > | Network | sparsity | Vgg19 | Resnet18 | Resnet50 | SqueezeNet | MobileNet v2 |
> > > | ---- | ---- | ---- | ---- | ---- | ---- | ---- |
> > > | Baseline  | dense | 71.41 | 77.19 | 78.60 | 71.01 | 68.99 |
> > > | Unstructured | 80 | 71.62 ($\pm$0.11) | 76.42 ($\pm$0.04) | 77.99 ($\pm$0.08) | 70.27 ($\pm$0.15) | 68.89 ($\pm$0.06) |
> > > | OVW          | 80 | 71.36 ($\pm$0.04) | 73.67 ($\pm$0.35) | 75.80 ($\pm$0.16) | 69.05 ($\pm$0.12) | 68.31 ($\pm$0.11) |
> > > | OVW permuted | 80 | 71.46 ($\pm$0.06) | 74.23 ($\pm$0.22) | 75.99 ($\pm$0.18) | 69.29 ($\pm$0.15) | 68.52 ($\pm$0.05) |
> > > | $\Delta$   | | +0.10 | +0.56 | +0.19 | +0.24 | +0.21 |
> > > | V  | | 64 | 64 | 64 | 44.48 | 32.02 |
> > > | Unstructured | 90 | 71.35 ($\pm$0.05)  | 74.87 ($\pm$0.07) | 78.02 ($\pm$0.04) | 70.29 ($\pm$0.16) | 68.40 ($\pm$0.19) |
> > > | OVW          | 90 | 71.29 ($\pm$0.03)  | 71.14 ($\pm$0.10) | 72.72 ($\pm$0.13) | 65.69 ($\pm$0.59) | 66.07 ($\pm$0.64) |
> > > | OVW permuted | 90 | 71.37 ($\pm$0.19)  | 72.25 ($\pm$0.15) | 73.26 ($\pm$0.22) | 65.80 ($\pm$0.73) | 67.46 ($\pm$1.03) |
> > > | $\Delta$   | | +0.08 | +1.11 | +0.54 | +0.11 | +1.39 |
> > > | V  | | 64 | 64 | 64 | 44.48 | 32.02 |

---

> ### Author Response · Authors · 2022-08-02
> **Response to Reviewer C8Jt part1**
>
> Thank you for reviewing our paper.
> In response to the constructive feedback, we have spent the past week revising our manuscript and conducting new experiments.
>
>
> **Writing**
> > Overall the text requires a significant amount of editing... I suggest the authors spend some time editing the text for both grammar and structure.
>
> We have revised typos addressed in figure 1, abstract, and Section 2.1 and carefully checked the remaining article. The readability of the paper will be improved if this paper is accepted.
>
> We also edit the article structure of the article. Detail modifications are listed as follows:
> + we reorganized the paragraph segmentations of Section 1 and 2.
> + We revise Section 3.2 to “The OVW pattern's Advantages in Convolution Computation”.
> + We revise paragraphs in Section 3.3 to cope with the Algorithm 1 illustration.
> + We merged Section 3.3 and 3.4.
> + We add brief abstract for Section 3 and  4.
> + We modify the paragraph segmentations in Section 4.2 to be more precise on the details.
> + We add more text in Section 5 and Section 6 to be more accurate on our experiment results and the limitations of our methods.
>
>
> > The text could also be more precise to help ensure reproducibility. Section 4.2 is a particular example of this, with vagues statements.
>
> Thanks for pointing it out. the key flaw of K-means is that it does not generate clusters of the same size as we required and we have to reassign some of the rows after that which hinders the clustering results. This is the main reason why the Balanced K-means algorithm was introduced.
> The following from revised section 4.2 briefly explains this algorithm:
> “In each iteration of balanced K-means, instead of assigning each data vector to its nearest cluster center as it is in the origin K-means algorithm, we formed a distance matrix between all the vector and current cluster center, minimizing the sum of distance while each cluster contains the same amount of data vector. This problem can be converted to a bipartite matching and Kuhn-Munkres algorithm is employed to solve it.”
>
> **Illustration**
> > I believe figure 1's illustration of "inter-vector-wise" sparsity contains a small mistake
>
> Thanks. We have revised typos addressed in figure 1 and updated the manuscript.
>
> >  Figure 2 is confusing to me.
>
> In Fig 2, we are trying to demonstrate how input data are organized and fetched into shared memory and which threads they are assigned to. The filter data stays in the sparse matrix format (i.e. CSR format) loading process. The Recover operation converts __row_idx__ to __ic, h, w__ indicating that the original structure of the filter is recovered. This process is done by __Cal_Thread_Offset__ function.
> We have modified Figure 2 for better demonstration. Flow instructors dealing with pointers and descriptors are changed to dotted arrows.

---

### Meta-Review · Area_Chair_kekX · 2022-08-28

**Recommendation:** Accept
**Confidence:** Less certain

**Metareview:**

While the reviewers had a difference of opinion in their scores of the paper, on balance I think the overall evaluation described in the text of the reviews leans strongly towards acceptance. This is especially the case because the only negative reviewer wrote "if these two changes are made (editing for language/grammar, and reporting confidence intervals in the main text) I would be willing to increase my score to accept" and I think that the authors have shown that they are going to do that in their updated version. The paper is well structured (if not so clearly written), and provides a novel approach to co-designed sparse convolution. It's really nice to see this sort of co-design, which is common in the ML systems space at the hardware-software level, being applied higher in the systems stack to achieve such significant speedups (even with not that much sparsity) for structured sparsity on existing commodity hardware. In general, when I see a technically strong paper where the weaknesses are presentation and spelling/grammar, I lean toward acceptance, with the understanding that the authors can incorporate the reviewers' feedback to improve the writing in the final version of their paper. Applying that reasoning here, I recommend acceptance.

**Award:**

No

---

### Decision · Program_Chairs · 2022-09-14

Accept